# Combinatorial Glycomic Analyses to Direct CAZyme Discovery for the Tailored Degradation of Canola Meal Non-Starch Dietary Polysaccharides

**DOI:** 10.3390/microorganisms8121888

**Published:** 2020-11-29

**Authors:** Kristin E. Low, Xiaohui Xing, Paul E. Moote, G. Douglas Inglis, Sivasankari Venketachalam, Michael G. Hahn, Marissa L. King, Catherine Y. Tétard-Jones, Darryl R. Jones, William G. T. Willats, Bogdan A. Slominski, D. Wade Abbott

**Affiliations:** 1Lethbridge Research and Development Centre, Agriculture and Agri-Food Canada, Lethbridge, AB T1J 4B1, Canada; kristin.low@canada.ca (K.E.L.); xiaohui.xing@canada.ca (X.X.); paul.moote@canada.ca (P.E.M.); douglas.inglis@canada.ca (G.D.I.); marissa.king@canada.ca (M.L.K.); darryl.jones@uleth.ca (D.R.J.); 2Department of Chemistry and Biochemistry, University of Lethbridge, Lethbridge, AB T1K 3M4, Canada; 3Department of Agricultural, Food & Nutritional Science, University of Alberta, Edmonton, AB T6G 2P5, Canada; 4Department of Biological Sciences, University of Lethbridge, Lethbridge, AB T1K 3M4, Canada; 5Complex Carbohydrate Research Center, University of Georgia, Athens, GA 30602, USA; sivav@uga.edu (S.V.); hahn@ccrc.uga.edu (M.G.H.); 6Center for Bioenergy Innovation, Oak Ridge National Laboratory, Oak Ridge, TN 37831, USA; 7Department of Plant Biology, University of Georgia, Athens, GA 30602, USA; 8School of Natural and Environmental Sciences, Newcastle University, Newcastle Upon Tyne NE1 7RU, UK; catherine.tetard-jones@newcastle.ac.uk (C.Y.T.-J.); william.willats@newcastle.ac.uk (W.G.T.W.); 9Department of Animal Science, University of Manitoba, Winnipeg, MB R3T 2N2, Canada; bogdan.slominski@umanitoba.ca

**Keywords:** canola meal, *Brassica napus* L., CAZymes, enzyme discovery, glycosidic linkage analysis, glycome profiling, plant cell wall, non-starch polysaccharides, intestinal microbiome

## Abstract

Canola meal (CM), the protein-rich by-product of canola oil extraction, has shown promise as an alternative feedstuff and protein supplement in poultry diets, yet its use has been limited due to the abundance of plant cell wall fibre, specifically non-starch polysaccharides (NSP) and lignin. The addition of exogenous enzymes to promote the digestion of CM NSP in chickens has potential to increase the metabolizable energy of CM. We isolated chicken cecal bacteria from a continuous-flow mini-bioreactor system and selected for those with the ability to metabolize CM NSP. Of 100 isolates identified, *Bacteroides* spp. and *Enterococcus* spp. were the most common species with these capabilities. To identify enzymes specifically for the digestion of CM NSP, we used a combination of glycomics techniques, including enzyme-linked immunosorbent assay characterization of the plant cell wall fractions, glycosidic linkage analysis (methylation-GC-MS analysis) of CM NSP and their fractions, bacterial growth profiles using minimal media supplemented with CM NSP, and the sequencing and *de novo* annotation of bacterial genomes of high-efficiency CM NSP utilizing bacteria. The SACCHARIS pipeline was used to select plant cell wall active enzymes for recombinant production and characterization. This approach represents a multidisciplinary innovation platform to bioprospect endogenous CAZymes from the intestinal microbiota of herbivorous and omnivorous animals which is adaptable to a variety of applications and dietary polysaccharides.

## 1. Introduction

Canola meal (CM) is the protein rich by-product generated by oil extraction from the seeds of several registered rapeseed (*Brassica napus* L.) canola cultivars specifically developed by Canadian plant breeders for the production of high-quality oil [1]. Recently, CM has emerged as a staple for dairy cattle production [2] and has also shown promise as an alternative feed for monogastric species, such as swine [3] and poultry [4,5,6]. Historically, the use of CM in poultry diets has been limited due to its correlation with weaker performance outcomes when compared to other protein sources, such as soybean meal (SBM), despite containing similar levels of protein (37% CM: 46% SBM) [7,8]. This outcome has been attributed to the higher abundance of plant cell wall-derived non-starch polysaccharides (NSP) (32% CM: 22% SBM) and lignin (10% CM: 3% SBM) in CM [8,9,10]. The use of enzymes in CM feed mixes at higher inclusion rates has been pursued in an attempt to increase the digestibility of CM by monogastrics with limited success [8,11,12].

The chicken intestinal microbiome, while not as efficient as that of the cattle rumen at carbohydrate metabolism, is replete with bacterial species well known for their ability to deconstruct plant cell walls and liberate metabolizable carbohydrates. Indeed, *Bacteroidetes* and *Firmicutes* dominate the phylotypes of both the chicken cecal [13] and gastrointestinal [14] microbiomes. Previous work reported that the carbohydrate active enzyme (CAZyme) content within the chicken cecal microbiome is enriched in select families of core enzymes (e.g., GH5 cellulase; GH10 xylanases) and devoid of others that possess complementary activities (e.g., GH6, GH7, GH45, and GH48 cellulases; GH11 xylanase; and GH12 xyloglucanase). This suggests that the chicken intestinal microbiome is tuned towards the digestion of specific dietary polysaccharides (e.g., starch), and that the metabolism of more complex substrates may be less efficient. Thus, introducing enzymes from other biological sources, such as the digestive organs of ruminants or other herbivorous species, or enriching the abundance of limiting enzyme activities found within the chicken microbiota may have utility for improving the digestion of alternative feedstuffs, such as CM.

Enzyme feed additives are attractive means to improve the digestibility of feedstuffs. Historically, they have focused mainly on corn and soybean meal (SBM) as the two dominant feed ingredients for monogastric animals. Treatment of SBM with proteases can inactivate anti-nutritional factors and increase nutrient accessibility [15]. High-starch diets can be more readily digested by poultry when supplemented with α-amylase, leading to increased weight gain [16]. Despite these reports, enzyme additives can have mixed results in poultry; the inclusion of β-glucanases in the diet has shown no improvement in growth performance of barley-fed chickens, despite targeting β-glucans dominant in barley cell walls [17]. Furthermore, non-specific enzymatic cocktails of proteases, amylases, and xylanases, while successful for barley- and wheat-based diets [18], have shown limited improvements to animal performance when used in conjunction with corn and SBM diets [19,20,21,22]. Specifically, the addition of exogenous enzymes has been shown to assist in the digestion of CM NSP, such as pectins and arabinoxylans [10,23], often using multi-enzyme mixtures [11]. CM NSP are more recalcitrant to digestion than those of SBM and corn [8] and thus these fibers represent a novel target substrates for enzyme assisted digestion of CM cell walls.

Very little research has been conducted to optimize the performance of enzymes for unique feed ingredients or within the digestive system of individual livestock species. To address these limitations, we have deployed an enzyme discovery innovation platform (Appendix A) that incorporates high-resolution characterization of complex substrates, isolation of chicken-associated bacteria using selective metabolism, and bioinformatic and biochemical approaches to mine bacterial genomes for candidate enzymes. For this study, the glycomes of cold-pressed (expelled) and pre-press solvent-extracted CM were characterized using two-different glycomic approaches. A preparative method for extraction of CM NSP was optimized to purify sufficient material to ascertain the impact of the NSP and CM on the broiler cecal bacterial community using a continuous-flow mini-bioreactor system, and to use the system to enrich for bacteria with CAZymes of interest. The genomes of six isolates were mined for CAZymes that are proficient in the hydrolysis of CM polysaccharides. Two enzymes with the potential to alleviate poor performance outcomes associated with CM digestion were recombinantly produced and characterized for their ability to digest intact and substrates identified within CM NSP. This study represents a proof of concept for a CAZyme discovery platform that can be tailored to mine the genomes of intestinal bacteria for enzymes active on structurally diverse plant cell wall material.

## 2. Materials and Methods

### 2.1. Extraction of NSP from CM

Large-scale NSP extraction was adapted from previous cell wall extraction methods specialized for oil-rich plant matter [24], in order to obtain sufficient quantities of NSP required for the isolation and selective growth of bacterial samples. Raw cold-pressed (Hartland Hutterite Colony, Badshaw, AB, Canada) or pre-press solvent-extracted (Matt A. Oryschak, University of Alberta, Edmonton, AB, Canada) CM was ball-milled to a fine powder, and 500 g was weighed and washed in 2.5 L of 100% hexanes to remove residual lipid contaminants for 2 h with robust stirring, three times. CM residues were then washed three times (once for 16 h, twice more for 2 h each) with 2.5 L of 100% ethyl acetate to remove residual hexane and other small molecule impurities. Ethyl acetate insoluble residues were extracted by the addition of 2.5 L of 80% (*v*/*v*) ethanol three times for 16 h each. Alcohol insoluble residues (AIRs) were washed for 20 min three times each in 2.5 L of 100% acetone, followed by triplicate washes in 100% methanol. The residue was vacuum-dried for approximately 3 days (or until dryness) at room temperature, and then ball-milled to a fine powder.

Destarching via enzymatic treatments was performed to obtain non-starch polysaccharides (NSP); 100 g of ball-milled CM AIR was weighed and suspended in 0.3 L of 100% DMSO and stirred gently at 80 °C for 1 h. Another 0.3 L of 100% DMSO was added and the mixture was stirred gently at 100 °C for a further 5 min. The solution temperature was lowered to 70 °C, and 0.9 L of amylase solution (90,000 U mL^−1^ thermostable *Bacillus licheniformis* α-amylase (Megazyme) in 100 mM sodium acetate pH 5.0, 5 mM CaCl_2_) was added with gentle stirring for 16 h. The temperature was then adjusted to 50 °C, and 0.6 L of 200 mM sodium acetate buffer pH 4.5 was added followed by an additional 0.6 L of 3 mg mL^−1^
*Aspergillus niger* amyloglucosidase (Megazyme) in 200 mM sodium acetate pH 4.5. The solution was stirred gently at 50 °C for 2 h. The resulting destarched residue suspension was then dialyzed (MWCO 3500 Da) extensively against deionized water. Samples were then removed from dialysis tubing, and water-insoluble CM NSP residues were separated, lyophilized to dryness, and ball-milled to a fine powder prior to subsequent use. Cold-pressed CM was used for the artificial intestine and broiler experiments, and all bacterial isolates were obtained from cold-pressed CM.

### 2.2. Antibody-Based Polysaccharide Characterization

Two monoclonal antibody-based screens were used to characterize the plant cell wall glycans of CM. The first enzyme-linked immunosorbent assay (ELISA) screen contains 155 plant cell wall glycan-directed monoclonal antibodies (mAbs) targeting a broad spectrum of epitopes [25,26]. Briefly, CM AIRs were sequentially extracted from CM whole cell wall using 50 mM ammonium oxalate, 50 mM sodium carbonate (with 0.5% (*w*/*v*) NaBH_4_), 1 M KOH (with 1% (*w*/*v*) NaBH_4_), 4 M KOH (with 1% (*w*/*v*) NaBH_4_), sodium chlorite (acidified by glacial acetic acid), and 4 M KOH (with 1% (*w*/*v*) NaBH_4_) resulting in AO, SC, 1 M KOH, 4 M KOH, CH, and 4 M KOHPC extracts, respectively [26]. Background signals resulting from wells carrying no antigen were subtracted from all readings, and the OD output of the ELISAs were plotted as heat maps. The glycome profiles are organized based on current information about the epitopes recognized by the mAbs [25,27,28,29,30,31]; antibodies that recognize similar epitopes are grouped together. The second, Micro Array Polymer Profiling (MAPP) screen probed differentially fractionated CM with 10 mAbs with well-established glycan epitope binding profiles [32]. CM cell wall glycans were sequentially extracted using 50 mM diamino-cyclo-hexane-tetra-acetic acid (CDTA) pH 7.5, 4 M NaOH (with 1% (*w*/*v*) NaBH_4_), and cellulase enzyme (CDTA, NaOH, and cellulase fractions respectively). These fractions were robotically printed (Arrayjet, Roslin, Scotland) onto 0.45 µM nitrocellulose membrane in 4 dilutions, creating microarrays, which were probed with mAbs as previously described [32]. Background signals from control arrays that were probed with secondary but not primary mAbs was subtracted from, and signals are reported as means of triplicates.

### 2.3. Polysaccharide Linkage Analyses

Cell wall of each CM sample was prepared as de-starched AIRs according to the previous report [24], with slight modifications; the de-starched sample remained in tubing after extensive dialysis against deionized water was centrifuged and the resulting precipitate was collected, lyophilized, and ball-milled.

Starting material (~0.3 mL mg^−1^) was subjected to sequential fractionation with the following solutions: 50 mM EDTA (pH 6.5), 50 mM Na_2_CO_3_ containing 25 mM sodium borodeuteride (NaBD_4_, 99 atom % D, Alfa Aesar), and 4 M KOH containing 25 mM NaBD_4_ for 24 h each at room temperature with gentle magnetic stirring, which was modified based on the literature [33]. After each extraction, the soluble fraction was collected by differential centrifugation followed by neutralization. The residue was washed with deionized water three times with centrifugation in between, and all washes were pooled into the corresponding soluble fraction. The EDTA and Na_2_CO_3_ extracts were pooled. Final fractions were dialyzed extensively against deionized water and lyophilized.

Alternatively, before fractionation the cell wall was pre-treated with 0.25 M NaBD_4_ solution (0.2 mL mg^−1^ of starting material) at 4 °C for 24 h [34], followed by neutralization by slow dropwise addition of 10% (*v*/*v*) acetic acid over ice and centrifugation (3000× *g*, 30 min). The resulting supernatant was pooled with three water washes of the pellet with centrifugation between washes, and the pooled solution was further incorporated into the EDTA and Na_2_CO_3_ fraction isolated from the washed pellet that was sequentially fractionated using the same method as described above.

Linkage analyses of the two cell walls (cold-pressed and solvent-extracted) and their fractions were performed by carboxyl reduction of uronic acids to their corresponding 6,6-dideuterio neutral sugars [35] followed by GC-MS analysis of partially methylated alditol acetate derivatives according to the published protocol [36], except that sample solution (1 μL) was splitless-injected to an Agilent 7890A-5977B GC-MS system (Agilent Technologies) installed with a Supelco SP-2380 column (30 m × 0.25 mm × 0.20 μm, Sigma-Aldrich) with a constant helium flow of 0.8 mL min^−1^ and with optimized oven temperature programmed to start at 55 °C (hold 1 min) followed by increasing at 30 °C min^−1^ to 135 °C, 3 °C min^−1^ to 180 °C (hold 20 min), 1.5 °C min^−1^ to 200 °C, and 3 °C min^−1^ to 255 °C (hold 3 min). Four separate experiments were conducted to each unfractionated cell wall, and three to each fraction.

### 2.4. Cecal Digesta Collection and Establishment of Bacterial Communties in Continuous-Flow Mini-Bioreactors

Digesta were harvested within 30 min of death from the ceca of three healthy ca. six-month-old mature male broiler breeders obtained from a local producer (i.e., humanely culled by the producer according to industry standards and transported to the Lethbridge Research and Development Centre immediately after euthanization). Ceca were removed and digesta were harvested in an anaerobic chamber containing a 85:5:10% N_2_:H_2_:CO_2_% atmosphere (referred to as the nitrogen atmosphere hereafter). Harvested cecal digesta were thoroughly mixed, and a subsample of the digesta was used for the direct isolation of bacteria. Subsamples of digesta were also used to establish a broiler cecal bacterial community in continuous-flow mini-bioreactors.

For cecal communities established in bioreactors, digesta were suspended in reduced BRM [37], and were added to individual bioreactor vessels at a final concentrate of 25% *w*/*v*. The mini-bioreactor array consisted of a multi-vessel continuous-flow system [37] that was situated in a Thermo Forma 1025 anaerobic chamber (Thermo Fisher Scientific Inc., Waltham, MA, USA) with a nitrogen atmosphere. Two 24-channel peristaltic pumps (205U, Watson Marlow, Concord, ON, Canada) with low flow capabilities provided nutrient exchange to individual vessels. Each vessel had an inflow and outflow tube, and peristaltic pumps were set for an intake rate of 2 RPM (3.76 mL/h) and an outtake rate of 4 RPM (7.52 mL/h). Cold-extracted CM (5.0%) or NSP extracted from CM (0.9%) was added to the BRM-digesta mixture in the vessels. A control treatment (CON) consisted of the BRM medium with cecal digesta not supplemented with either NSP or CM. Flow to/from the bioreactor vessels was commenced after 24 h after the addition of the cecal digesta; inflow BRM was maintained in individual 1 L media bottles, and the medium was replaced daily. The volume of each vessel was maintained at 60 mL, with four replicate vessels per treatment. A micro stir bar (10 mm × 1.5 mm) was placed in each vessel, and vessels were placed on a multi-position stir plate (Variomag^®^ POLY 15, Thermo Fisher Scientific Inc., Waltham, MA, USA) set at 200 rpm to ensure consistent mixing of the vessel contents. The ambient temperature was maintained at 37 °C with top-mounted heaters. The temperature was verified using digital thermometers positioned throughout the chamber. Moisture, oxygen, and sulfurous gases were minimized within the chamber using desiccant, palladium, and charcoal catalysts, and were replaced as needed (depending on moisture). Each vessel was fitted with sampling port, and samples were removed via a rubber septum using a sterile needle fitted to a 5 mL syringe.

Subsamples were collected from individual bioreactor vessels at day 0, 1, 2, 3, 4, 6, 8, and 10 using a 22G 7.6 cm hypodermic needle (Air-Tite Products Co., Inc., Virginia Beach, VA, USA) fitted to a 5 mL syringe. To collect the sample, the sterile needle was inserted through the septum of the sampling port of the vessel.

### 2.5. Temporal Characterization of Bacterial Communities in Mini-Bioreactor Vessels

On all eight sample days, 1 mL of the contents from each vessel was placed into individual sterile 2 mL screw-capped tubes, tubes were centrifuged (13,200× *g* for 10 min), the supernatants were transferred to new screw-capped tubes, and both samples were flash frozen in liquid nitrogen. Samples were stored at −80 °C until processed. The frozen pellet was re-suspended in 200 µL of lysis buffer containing 10 mg mL^−1^ lysozyme, and DNA was extracted from using the DNeasy Blood and Tissue kit (Gram Positive protocol; Qiagen Inc., Hilden, Germany). DNA from the four replicate bioreactors per treatment was pooled by volume for Illumina sequencing. Extracted DNA was quantified using a Qubit 4 fluorometer (Thermo Fisher Scientific Inc., Waltham, MA, USA), and PCR was amplified for 16S rRNA gene metagenomics analyses using the protocol developed by Kozich et al. [38], which targets the V4 region of the 16S rRNA gene with a dual-indexing strategy. The PCR mastermix included 12.5 µL of Paq5000 Hi Fidelity Taq Master Mix (Agilent Technologies, Mississauga, ON, Canada), 1 µL of 10 µM of forward primer (V4-Read 1 Primer TATGGTAATTGTGTGCCAGCMGCCGCGGTAA), 1 µL of 10 µM of reverse primer (V4-Read 2 Primer AGTCAGTCAGCCGGACTACHVGGGTWTCTAAT) (Integrated DNA Technologies, Coralville, IA, USA), 8.5 µL of Nuclease free water (Qiagen Inc., Hilden, Germany), and 2 µL of genomic DNA template (30–50 ng). Reactions were amplified on a thermocycler (Mastercycler Pro S; Eppendorf, Mississauga, ON, Canada) using the following conditions: 95 °C for 2 min; 25 cycles of 95 °C for 20 s, 55 °C for 15 s, and 72 °C for 2 min; and a final elongation cycle at 72 °C for 10 min. Amplicons were purified with AMPure XP beads (Beckman Coulter Diagnostics, Brea, CA, USA), and checked for quality and size with a Bioanalyzer 2100 (Agilent Technologies, Mississauga, ON, Canada). Quantification of the amplicons was done using a Qubit 4. Samples were normalized to 4 nM, pooled, denatured with NaOH, and further diluted with HT1 (Illumina, San Diego, CA, USA) to produce a 4 pM library for analysis using a MiSeq System (Illumina, San Diego, CA, USA). Twenty percent PhiX control DNA was added to the library as a sequencing control. The library was loaded using a MiSeq Reagent Kit v2 500-cycle, and run on an Illumina MiSeq platform (Illumina, San Diego, CA, USA). The Q30 score for the output data was 89%. QIIME2 [39] was used to process and classify bacterial reads. Raw reads were de-noised with DADA2 [40], and representative sequences and amplicon sequence variants (ASVs) were generated. Samples with a read depth of less than 22,000 were removed from the analyses. A phylogenetic tree of ASV sequences was generated, and the taxonomy of each ASV was identified by using a machine learning classifier pre-trained with the reference SILVA 132 database (lva-132-99-515-806-nb-classifier.qza). Evenness (i.e., Pielou’s evenness index) and α-diversity (i.e., Shannon’s index) were calculated. In addition, β-diversity was determined using unweighted UniFrac, weighted UniFrac, Bray-Curtis, and Jaccard distance/dissimilarity.

### 2.6. Selective Isolation of Intestinal Bacteria

Reduced samples from the vessels on days 3 and 10 were immediately transferred to an anaerobic chamber (Coy Lab Products) with a nitrogen atmosphere. Samples were diluted serially (10^0^–10^6^) in sterile, anaerobic 1X phosphate-buffered saline (PBS); 100 µL of each serial dilution was plated onto nutrient restricted media to select for microorganisms that could metabolise CM and CM NSP: Agar (1.5%) plates supplemented with minimalized medium (MM) and CM AIR (5%) or CM NSP (1%) as a sole carbohydrate source. The MM was adapted from [41]; a 1X MM solution contained the following: 0.23 g L^−1^ K_2_HPO_4_, 0.23 g L^−1^ KH_2_PO_4_, 0.23 g L^−1^ (NH_4_)_2_SO_4_, 0.46 g L^−1^ NaCl, 0.09 g L^−1^ MgSO_4_·7H_2_O, 40 mg L^−1^ CaCl_2_·2H_2_O, 2 g L^−1^ tryptone, 1 mg L^−1^ hemin, 4 g L^−1^ Na_2_CO_3_, and 0.5 g L^−1^ cysteine-HCl. Colonies were selected after 4 days of growth and re-streaked twice to obtain single colonies, numbered consecutively from 1 to 332, on agar (1.5%) plates containing Columbia medium supplemented with 1 mg L^−1^ hemin.

### 2.7. Identification of Bacterial Isolates

Liquid cultures were grown of all selected CM NSP-degrading single colony bacterial isolates in Columbia medium (Difco) supplemented with 1 mg L^−1^ hemin at 37 °C in an anaerobic chamber (Coy Lab Products, Grass Lake, MI, USA) in a nitrogen atmosphere. Bacterial cells were harvested by centrifugation at 8000× *g* for 10 min. Genomic DNA samples were extracted using a 96-well Bacteria Genomic DNA Miniprep Kit (BioBasic Inc., Markham, ON, Canada) and sent for 16S rDNA sequencing (Eurofins Genomics, Toronto, ON, Canada) using degenerate 27F and 1492R primers [42,43] to identify bacterial species.

### 2.8. Evaluation of Growth Profiency of Isolates on CM NSP

Growth proficiency of *Bacteroides* isolates was evaluated at 37 °C in an anaerobic chamber (Coy Lab Products, Grass Lake, MI, USA) in an atmosphere consisting of 85% N_2_, 10% CO_2_, and 5% H_2_. Canola meal NSP-utilizing numbered isolates (CMU#) corresponding to *Bacteroides* spp., CMU13, CMU19, CMU33, CMU36, CMU103, CMU108, CMU128, were cultured overnight in Columbia medium (supplemented with 1 mg L^−1^ hemin). Isolates were then subcultured overnight at 37 °C into MM supplemented with D-glucose (0.5%) to adapt to nutrient restricted media. Liquid culture growths were inoculated from overnight cultures and performed in triplicate, in MM supplemented with CM NSP (1%) or D-glucose (0.5%) as a control. NSP were insoluble which prevented us from using an anaerobic multiplexed robotic plate reader. Thus, growth profiles were assessed by OD600 nm using manual time point readings at 0, 5, and 24 h, with gentle manual agitation after each time point to promote NSP accessibility and bacterial growth.

### 2.9. Whole Genome Sequencing and CAZyme Identification

Six *Bacteroides* isolates, two from each species, were selected for whole genome sequencing using Illumina HiSeq4000 PE150 (*Bacteroides thetaiotaomicron* (*theta*) CMU13 and CMU108, *Bacteroides ovatus* CMU19 and CMU33, and *Bacteroides fragilis* CM103) or Illumina NovaSeq 6000 S4 PE150 (*Bacteroides fragilis* CMU36) (Genome Québec, Montréal, QC, Canada). Contigs were assembled de novo using the SPAdes assembler [44], and sequencing quality was assessed using QUAST [45]. Genomes were submitted to NCBI GenBank, with contigs smaller than 200 bp deleted prior to submission, and accession numbers/BioSample IDs are as follows: CMU13 (SAMN15915567), CMU108 (SAMN15915568), CMU19 (SAMN15915569), CMU33 (SAMN15915570), CMU36 (SAMN15915571), and CMU103 (SAMN15915572). Genomes were analyzed by Prokka [46] to annotate putative proteins.

### 2.10. CAZyme Phylogenetic Analyses and Target Selection

Prokka results were analyzed locally by dbCAN [47] to determine total CAZyme content (CAZome) for all isolated bacterial species. CAZyme families were chosen based on polysaccharide composition as determined by the glycomics analyses; predicted CAZyme genes from GH28, GH43, GH78, and CE6 families, active on rhamnogalacturonan-I (RG-I) backbone, arabinoxylan (AX), arabinan (AB), and homogalacturonan (HG), were selected from CMU13 and CMU19 isolates analyzed by SACCHARIS [48] to determine phylogenetic relatedness to biochemically characterized sequences. Sequences and accession numbers for characterized proteins were extracted from the CAZy database [49], and all protein sequences were aligned using MUSCLE [50]. ProtTest [51] was used for best-fit model selection using the MUSCLE sequence alignment, and FastTree [52] was used to generate the trees. Phylogenetic trees were annotated using iTOL [53]. Target enzymes were chosen based on distally related sequences, represented by novel clades within the phylogenetic trees. Domain boundaries were manually curated via predictions by dbCAN [54] and InterProScan [55].

### 2.11. Target CAZyme Gene Synthesis and Expression in E. coli

Selected gene targets (with GenBank-assigned locus tags in parentheses): N872 (IAG16_04270), N1089 (IAG16_05330), N2073 (IAG16_10110), N2589 (IAG16_12615), N3394 (IAG16_16585), K696 (IAG19_03410), K2605 (IAG19_12770), and K3550 (IAG19_17440) were codon optimized for expression within *E. coli* and gene synthesized as full-length constructs without their native signal peptide but including flanking *NdeI* and *XhoI* restriction sites at the 5′ and 3′, respectively (BioBasic Inc.). Genes were ordered synthesized in the pET24b vector for C-terminal His_6_-tag protein expression. Each protein construct vector was transformed into *E. coli* BL21 (DE3) Star cells (Thermo Fisher). Cells were grown in LB Miller broth containing 50 µg mL^−1^ kanamycin at 37 °C to an OD600 nm of 0.6, when protein expression was then induced by the addition of IPTG to a final concentration of 1 mM. The cell culture was incubated at 18 °C for 16 h prior to being harvested by centrifugation at 6500× *g* for 20 min at 4 °C. Cell pellets were stored at −20 °C until needed.

### 2.12. Recombinant Gene Expression and Purification

The cell pellet from 1 L of bacterial culture was thawed and resuspended in 50 mL of lysis buffer (20 mM Tris pH 8.0, 500 mM NaCl, 0.1 mg mL^−1^ lysozyme). Cells were homogenized by sonication for 2 min of 1 s intervals of medium intensity sonic pulses at a power setting of 4.5 (Heat Systems Ultrasonics Model W-225 and probe). Cellular debris was removed by centrifugation at 17,500× *g* for 45 min at 4 °C and passed through a 0.45 µm filter. The filtrate was loaded onto Ni-NTA resin and purified by immobilized metal affinity chromatography, whereby recombinant protein was eluted by an increasing gradient 0–500 mM imidazole in 20 mM Tris pH 8.0 and 500 mM NaCl. Protein was concentrated by centrifugation with 10 kDa (K2605) or 30 kDa (N1089) cutoff Amicon Ultra centrifugal concentrators (Millipore Sigma). His_6_-tagged protein was further purified using a HiLoad 16/60 Superdex 200 prep-grade size exclusion column (GE Healthcare, Chicago, IL, USA) in 20 mM Tris pH 7.5, 500 mM NaCl, 2% glycerol. Pure protein fractions were pooled and concentrated. Protein purification was monitored throughout by SDS-PAGE. All recombinant proteins were used freshly prepared for downstream enzyme activity assays.

### 2.13. pNP-Sugar Colourimetric Activity Assay

Enzyme activities were assayed against several *p*-nitrophenol (pNP)-sugar conjugate substrates: pNP-α-D-Man*p*, pNP-α-L-Ara*f*, pNP-α-L-Ara*p*, pNP-β-D-Xyl*p*, pNP-β-D-Gal*p*, pNP-β-D-Glc*p*, and pNP-acetate. Reactions contained 1 µM enzyme in 50 mM Tris pH 8.0 and were initiated via the addition of a final concentration of 1 mM pNP-sugar conjugate. Assays were carried out in triplicate in 96-well microtitre plates. Product release was measured by monitoring at OD405 nm (BioTek Instruments) every 1 min for 30 min. Final product concentration was calculated using a calibration curve for the hydrolysis product pNP. The final concentration of DMSO in each reaction did not exceed 2%. Background hydrolysis rates were measured and subtracted from reaction rates.

### 2.14. Thin Layer Chromatography (TLC)

Purified CAZymes were screened for activity on CM AIR and extracted CM NSP. Additionally, commercial sources of polysaccharides and synthetic substrates consistent with substrates identified within CM were tested, including: Potato RG-I (Megazyme), citrus PGA (Megazyme), wheat arabinoxylan (Megazyme), galacturonic acid ladder of mono, di, and triose forms (GalA_1,2,3_) (Megazyme), 3^2^-α-L-arabinofuranosyl-xylobiose (A^3^X) (Megazyme), 2^3^,3^3^-di-α-l-arabinofuranosyl-xylotriose (A^2,3^XX) (Megazyme). Reactions contained 1 µM enzyme and 5 mg mL^−1^ substrate in 20 mM Tris pH 8.0 and incubated overnight at 37 °C with mild shaking to maximize accessibility of insoluble substrates (i.e., CM AIR and CM NSP). After incubation, the samples were heat treated at 100 °C for 10 min to denature the enzyme and terminate the reaction. Samples were then shortly centrifuged at 8000× *g* to pellet denatured protein and undigested insoluble substrate particles from the product. Digested samples were spotted (total 6 µL; spotted 3 times with 2 µL each) onto TLC plates (TLC Silica Gel 60; EMD Millipore). The samples were dried between multiple rounds of spotting. Appropriate monosaccharide standards were included as controls (6 µL of 1 mM concentration); D-galactose (D-Gal), L-arabinose (L-Ara), D-xylose (D-Xyl), D-rhamnose (D-Rha), D-glucose (D-Glc), and D-mannose (D-Man). The samples were resolved using a mobile phase of 2:1:1 butanol:acetic acid:H_2_O, and dried prior to visualization with an orcinol solution (70:3, acetic acid:sulfuric acid with 1% orcinol) and heating at 100 °C for 3–5 min.

## 3. Results

### 3.1. Whole Plant Cell Wall Characterization via Antibody-Based Glycome Profiling

While the monosaccharide composition of CM has been elucidated [8], defined polysaccharide content and linkages within CM NSP remained largely unknown [56]. Enzyme discovery for targeted biomass deconstruction is quite difficult without knowledge of the structure of CM NSP. However, determination of polysaccharide composition in cell wall rich samples such as CM is challenging because of the sheer complexity of structures present and the variety of molecular associations between them. Biochemical approaches enable monosaccharides and glycosidic linkages to be quantified, but these cannot always be assigned with confidence to particular polysaccharides because of the polymer deconstruction that is required for analysis. In contrast, although mAb based techniques such as ELISAs and MAPP are only semi-quantitative, they are based on the extraction of largely intact polysaccharides. Moreover, the glycan epitopes recognized by mAbs typically encompass several linked monosaccharides, and can be highly specific for a particular polysaccharide. Thus, ELISAs and MAPP provide useful information about polysaccharides *per se*, and changes to those polysaccharides due to enzyme processing. 

CM plant cell wall fractions were extracted and analyzed for their glycome profiles using 155 plant cell wall glycan-directed mAbs [25,26] (Figure 1A). These results determined that CM contains all major groups of non-cellulosic polysaccharides typically found in higher plant cell walls, including xyloglucans, xylans, rhamnogalacturonans (RG), and galactans. Mannans were notably absent from the CM, and arabinogalactans were also not as abundant as seen in some other plant cell wall preparations (Figure 1A). Additionally, MAPP was used to screen a targeted panel of glycan-directed mAbs (Figure 1B), presenting results consistent with the broad-panel mAbs; CM AIR and NSP contain glycans consistent with those found in pectins (including RG-I and -II), xyloglucan, and xylan (Figure 1C).

### 3.2. Linkage Analyses of CM NSP

The linkage analyses of cell walls of cold-pressed and solvent-extracted canola meal were conducted for the first time. Sequential fractionation by extractants with increasing strength was conducted to the cell walls with and without the pre-treatment of NaBD_4_. The reducing agent NaBH_4_ acts on the reducing ends of polysaccharides and on the carboxyl groups resulting from alkaline cleavage of the esters of uronic acids (e.g., the methyl ester of galacturonic acids in pectins and ester link of the glucuronic acids of the hemicellulose glucuronoxylan) to prevent oxidative degradation during alkaline extraction [58,59,60]. NaBD_4_, a deuterated form of NaBH_4_, was used here for the analytical purpose of deuterium labelling changes that can be tracked by GC-MS analysis [35]. As expected, a much higher yield of the EDTA and Na_2_CO_3_ extract (32.1% versus 16.4%) and a much lower yield of the 4 M KOH fraction (29.0% versus 48.4%) was found in the NaBD_4_ pretreated cold-pressed sample than the untreated one, while slight difference in the insoluble residue yield (38.9% versus 35.2%) between the two was observed (Appendix A), indicating that NaBD_4_ treatment greatly increased the extractability of polysaccharides from the cell wall and enabled the release of more polysaccharides by the relatively weaker extractants EDTA and Na_2_CO_3_. Similar trends were observed in the solvent-extracted CM fractions.

The cell walls and their fractions were subjected to carboxyl reduction-methylation-GC-MS analysis for the composition of linkages from uronic acids and neutral sugars [36]. Results showed that 4-Glc*p* and 4-GalA*p* were the two most abundant linkages in the whole cell wall (Figure 2, Appendix A, Appendix A) indicating higher abundance of cellulose and pectins than hemicelluloses in the sample. The 4-GalA*p* from pectins and 4-Glc*p* from cellulose were found dominant in the pooled EDTA + Na_2_CO_3_ fraction and the alkali insoluble residue, respectively, and high amounts of linkages (e.g., 4-Xyl*p*) typical for hemicelluloses were observed in the 4 M KOH fraction, which was in agreement with the previous reports on the fractionation of higher plant cell wall [33,61].

Based on the linkage data (Appendix A), the relative composition of polysaccharides were estimated by assigning glycosidic linkages to polysaccharides (Table 1) and summing up the compositions of assigned linkages for corresponding polysaccharides following the published protocol [36], with the modifications that t-Fuc*p* was additionally assigned to arabinogalactan-II (AG-II) according to the recent reports [62,63] and extensin was grouped with AG-II based on the finding of the presence of extensin in the cell wall by the monoclonal antibody assay (Figure 1B) and the consideration that AG-II structure contains all the glycosidic linkage types of extensin glycoprotein [63]; 3-Glc*p* was assigned to callose instead of the mixed-linkage (1,3;1,4)-β-D-glucans that dicotyledonous plants lack [36]. Results showed the cell wall was rich in cellulose, pectins (homogalacturonan (HG), RG-I), arabinan (AB), and xyloglucan (XG) (Figure 3, Appendix A). It is well accepted that linear 5-linked AB are most prevalent in the side chains of RG-I [64,65,66], although they can also be found as part of arabinogalactan glycoproteins (AGPs). Considerable amounts of AG-I and AG-II structures were also found in the cell wall, which could be present as side chain structures of RG-I [64,65,66] or attached to AGPs. The possible crossing-linking between AB, AG-I and AG-II in the cell wall was supported by the observation of co-extraction and enrichment of them with the HG and RG-I in the pectin-rich EDTA and Na_2_CO_3_ fraction; however, direct evidence needs to be found in future studies in order to confirm the existence of such structures in the canola meal sample. XG were the most abundant hemicellulose in the canola meal cell wall, followed by heteroxylan (HX) and heteromannan (HM) in decreasing abundance (Figure 3). It was apparent that both the XG and HX were enriched in the strong alkali soluble fraction, which was in good agreement with previous reports [33,61], indicating both XG and HX were intensively cross-linked to the cell wall matrix and therefore strong extractants are needed to break the bonding and release them.

### 3.3. Temporal Characterization of Broiler Cecal Bacterial Communities in Mini-Bioreactors

There was no difference in α-diversity (*p* ≥ 0.298) or evenness (*p* ≥ 0.418) of communities within the mini-bioreactor vessels among the CM, NSP, and control treatments over time. However, a reduction in α-diversity was observed within the first 24 h for all treatments; Shannon’s Index values ranged from 6.7 to 6.9 at day 0, and from 4.2 to 4.6 at day 1. From day 1 onward, bacterial diversity was stable (*p* ≥ 0.083), ranging from 3.3 to 4.9 over the remaining nine days of the experiment. Despite the loss of diversity, bacteria in the phyla *Firmicutes* and *Bacteroidetes* were retained, although selection for some taxa occurred over time independent of treatment (Appendix A). In this regard, bacteria in the families *Bacteroidaceae* and *Fusobacteriaceae* increased over time. The β-diversity (i.e., structure) of bacterial communities was not altered by NSP relative to the control treatment (*p* ≥ 0.441). In contrast, the structure of communities was altered for the CM treatment as determined by unweighted UniFrac (*p* = 0.045) and Jaccard (*p* = 0.055) (Figure 4). However, no or modest quantitative differences in β-diversity were observed among the CM and CON treatments as determined by weighted UniFrac (*p* = 0.237) and Bray-Curtis (*p* = 0.067). The observed impact on β-diversity for the CM treatment was attributed in part to selection for bacteria within the *Veillonellaceae* family (Appendix A).

### 3.4. Enrichment and Isolation of CM NSP-Degrading Bacterial Species

Bacteria were isolated directly from cecal digesta or from the bioreactors (days 3 and 10); 332 isolates able to grow on an agar medium supplemented with MM and CM or CM NSP as a carbohydrate source were recovered. Of the 332 isolates, 325 were culturable in liquid rich media. We identified 100 intestinal bacterial isolates with the ability to degrade and metabolize CM NSP via growth on CM NSP-supplemented agar plates, and subsequent species identification was performed via 16S rDNA sequencing using degenerate 27F and 1492R primers [42,43] (Figure 5A). *Enterococcus* spp., and *Bacteroides* spp. were the most predominant bacteria able to utilize CM NSP.

### 3.5. Growth Metrics of “High-Efficiency” CM Degraders

Based on the bioreactor community analyses, growth in liquid culture for the *Bacteroides* isolates were evaluated to identify “high-efficiency” utilizers (Figure 5B). Of the seven isolates, two *B. theta* CM NSP-utilizing isolates (CMU13 and CMU108) and two *B. ovatus* isolates (CMU19 and CMU33) were capable of moderate growth in CM NSP after 24 h (termed high-efficiency utilizers for our purposes), whereas three *B. fragilis* isolates (CMU36, CMU103, and CMU128) grew to low density.

### 3.6. Total CAZyme Content of Bacteroides CM-Degrading Isolates

Many members of *Bacteroidetes* are polysaccharide generalists that consume diverse cell wall polysaccharides. Therefore, the six top performing *B. theta* and *B. ovatus* isolates were selected for genome sequencing and prospecting (Table 2). Functional annotation of assembled contigs and putative protein identification was done using Prokka [46].

Prokka results were analyzed by dbCAN [54] to annotate predicted CAZomes for all seven isolated bacterial species (Table 3 and Table 4, and Appendix A). Significantly, *B. theta* and *B. ovatus* high-efficiency utilizers contain 2X and 2.1X, respectively, the total CAZyme content as the *B. fragilis* isolates, which predominantly results from the number of glycoside hydrolase family members. When comparing CM-metabolizing isolates to environmental “wild-type” strains such as *B. theta* VPI-5482 and *B. ovatus* ATCC8483, CM isolates contain markedly higher numbers of CAZyme genes in many families relevant to debranching and degrading plant cell wall polysaccharides and their associated downstream oligosaccharide products (Table 3 and Table 4, highlighted). Many of the enzymes identified, including those of GH28, GH43, GH78, and CE6 families, are especially relevant in the context of the ELISA- and MAPP-based screening results and the polysaccharides identified in CM. As these predictions are based upon genome sequence, future transcriptomic studies would be useful to quantify expression of genes under CM NSP conditions.

### 3.7. Target Selection for CAZymes with Potential Enzymatic Activity on CM NSP

Based on polysaccharide linkage analyses, genes from CAZyme families of interest were selected from the genomes of high growers and analyzed using our in-house pipeline SACCHARIS [48]. Gene sequences from bacterial isolates CMU13 (*B. theta*) and CMU19 (*B. ovatus*) for GH28 (pectins), GH43 (arabinoxylans), GH78 (RG-I/II), and CE6 (xylans) enzymes were embedded into phylogenetic trees comprised of characterized enzyme sequences (Figure 6). From these trees, we can predict enzymatic function with higher accuracy. Overall, the majority of GH43 enzymes were found to be highly related both to each other, when comparing those from isolate CMU13 to isolate CMU19, as well as to other characterized enzymes; few isolate gene sequences are in novel clades, with two most distantly related groupings identified (Figure 6). GH28, GH78, and CE6 family members have individual outliers, signifying targets that may possess unique enzyme specificities. A total of 8 CAZyme sequences from either CMU13 or CMU19 were chosen from GH28 (N2073, K2605), GH43 (N1089, K696), GH78 (N872, N3394, K3550), and CE6 (N2589) families based on these criteria (Figure 6).

The selected targets were further analyzed by InterProScan [55] to identify any accessory modules of interest (Figure 6). Five of the eight enzymes selected were determined to be multimodular. Of note, the CMU13 GH28 CAZyme N2073 contains a putative carbohydrate lyase domain that was not identified by dbCAN; and CMU13 N1089 contains two tandem GH43 subfamily 2 domains, only one of which is divergent from other GH43 family members in the phylogenetic tree.

### 3.8. Characterization of Novel CM-Targeted CAZymes

Full-length constructs of the selected enzyme targets were cloned for recombinant expression in *E. coli*, and evaluated by small-scale expression tests to determine expression conditions that result in large amounts of soluble protein. Of the eight enzyme targets selected, two full-length enzymes were found to produce soluble recombinant protein: CMU13 N1089, with two tandem GH43 domains; and CMU19 K2605, one GH28 CAZyme.

In order to screen CAZymes for function, colourimetric pNP assays were used. When pNP-sugar conjugate substrates are cleaved by enzymes with the required activity, the products can be detected and quantified spectrophotometrically. The GH43 family enzyme N1089 can function as both an arabinofuranosidase and xylosidase, consistent with other GH43 enzymatic activities (Figure 7A), whereas the GH28 enzyme K2605 demonstrated faint activity on pNP-galactose (data not shown), with no detectable activity on other pNP-sugar substrates.

pNP-conjugate substrates provide an initial activity-based screen for enzymatic function, however these small-molecules do not reflect the complexity within native polysaccharide substrates. Thus, purified N1089 and K2605 were screened for activity on CM AIR and extracted CM NSP. Additionally, numerous commercial sources of polysaccharides and synthetic substrates were tested against general activities. The GH43 enzyme N0189 was found to be active on CM AIR, as smaller products consistent with oligosaccharide production can be seen by TLC (Figure 7B). A similar cleavage product cannot be seen on CM NSP, likely due to the highly insoluble nature of the NSP substrate. No other oligo/polysaccharide-based substrates tested were shown to be cleaved by either N1089 or K2605, including: RG-I, polygalacturonic acid (PGA), wheat arabinoxylan, 2^3^-α-L-arabinofuranosyl-xylotriose (A^2^XX), arabinobiose (Ara_2_), arabinotriose (Ara_3_) (data not shown).

## 4. Discussion

Previous research and development of commercial enzyme products has shown that the addition of exogenous enzymes to animal feed can improve the digestion of NSP [67,68,69]; however, these approaches did not optimize the enzymes for unique structural features within the cell walls of feed or to function within the intestine of animals. NSP content (cellulose and non-cellulosic) varies greatly between common feeds in poultry diets [70]. Cereal grains such as corn, wheat, and barley are rich in arabinoxylans and β-glucans, whereas the protein-rich dietary components, soybean and canola meals, are also rich in xyloglucans, galactans, and pectic polysaccharides, in addition to the xylans. Here, we have used an enzyme discovery platform to identify enzymes within the genomes of cecal bacteria isolated from chickens. Importantly, tailored enzyme discovery required the development of a comprehensive analytic pipeline to fully characterize the structure of the substrate and to select for relevant enzyme activities. Accordingly, we conducted the first in depth glycomic analysis of CM and CM NSP cell walls through a holistic strategy involving glycome profiling and glycosidic linkage analyses.

Canola is the registered name for the rapeseeds of several cultivars of *Brassica napus* species, and CM is superior to its predecessor rapeseed meal (RSM) in nutritional values [1]. Previously, CMs from different varieties and lines were found to contain 8–10% sucrose, 2–3% oligosaccharides, 20–22% NSP, and 5–8% lignin and polyphenols. It was reported that CM was rich in cellulose, hemicelluloses, and pectins based on neutral sugar analysis using GC and total uronic acid colorimetric assay [56]. There have been very few reports on the structural characterization of cell wall polysaccharides of either RSM or the closest genetic relative to canola meal, *Brassica campestris* [71]. Oil-free de-hulled RSM contains arabinogalactan (1%), arabinan (2%), pectin (14.5%), cellulosic residue (7%), and small amounts of xylans [72]. The de-fatted seed meal of *B. campestris* is rich in AB, RG-I, AGPs, XGs, and xylans [73]. Both RSM and *B. campestris* seed meals have similar compositions with respect to pectins, AG, AB, glucuronoxylans, and XG with minor differences in some of the detailed structural features of AB and XG [71]. Linkage analysis was used in the previous studies, but uronic acids were not treated by carboxyl reduction before methylation analysis, resulting in the detection of only neutral sugar linkages. Notably, pectins were observed in several sequential extracts even in the final residue after alkaline extraction indicating a rigid matrix of the seed meals, suggesting that pre-treatments are needed to loosen the cell wall structure for increased digestibility by monogastric animals [71], further supported by an in vivo study [74]. 

We are not aware of any structural studies of the cell wall composition of CM beyond its monosaccharide composition and inferred polysaccharide content [56]. The first step of conventional monosaccharide composition analysis is acid hydrolysis or methanolysis of polysaccharides to their constituent monosaccharides, and it is widely accepted that this initial hydrolysis can yield an incomplete release of monosaccharides from some cell wall polysaccharides (e.g., the release of glucose from insoluble crystalline cellulose or galacturonic acids from pectins) and degradation of released monosaccharides [75,76]. Unlike direct hydrolysis of underivatized samples, per-methylated polysaccharides can be completely hydrolyzed by relatively mild acid such as 2 M TFA without signs of degradation [36,77]. In the current study, uronic acids in the CM cell walls were carboxyl-reduced to neutral sugars with deuterium labelling, followed by methylation-GC-MS analysis of the carboxyl reduced samples, which enabled not only complete hydrolysis of the insoluble pectin-rich cell walls to monosaccharides, but also the determination of linkage patterns of the released monosaccharides. Based on the linkage composition data, estimation of polysaccharide composition was performed following a published protocol [36]. Glycosidic linkage analysis determined that CM NSP are rich in pectins (RG-I, AB, galactans, HG, and arabinogalactans), XG, and AX (Figure 3). In addition to linkage analysis, the CM cell walls were comprehensively characterized using two glycome profiling methods, one based on ELISAs using a large and diverse collection of mAbs, the other based on spotted arrays, using a smaller set of mAbs. The ELISA-based glycome profiling showed the presence of non-cellulosic polysaccharides typical of higher plant cell wall preparations (xyloglucans, xylans, pectins, galactans and arabinogalactans), although mannan epitopes were largely absent, and arabinogalactan epitopes appear underrepresented compared with other plant cell walls (Figure 1A). The array-based glycome profiling also revealed the prominent presence of xyloglucans and arabinogalactans, with a lesser amount of a galactoglucomannan epitope (Figure 1B). However, the spotted array did not detect appreciable levels of HG, in contrast to the ELISA-based results, where HG epitopes were easily detected. Interestingly, destarching appears to modify CM cell walls, as evidenced by lower detection of xyloglucan (LM15) and AGP (LM2) epitopes, and slightly higher detection of the mannan epitope recognized by LM21 (Figure 1B). Amylase has been previously shown to extract NSP [78]. Both glycome profiling methods reveal the polymeric complexity of NSP prepared from CM, illustrating a highly diverse set of epitope structures within each of several polysaccharide classes. In addition, the glycome profiling revealed that xylans and xyloglucans, as well as some RG-I and AGP epitopes are quite tightly bound to the cell wall matrix and require quite harsh conditions to release them from the wall. Homogalacturonans appear to be less tightly integrated into the CM walls, since the majority of these epitopes are released prior to the chlorite step in the wall extraction sequence.

High-resolution methods glycome profiling and glycosidic linkage analyses used herein have now revealed the true complexity of NSP within CM (Figure 1), and the digestion of these complex dietary fibres is likely to require the addition of appropriate enzymatic activities for their digestion. Some candidate types of enzymes are suggested by the data presented here. Bi-functional enzymes offer a more comprehensive breakdown strategy encoded within a single polypeptide, where the presence of multiple CAZyme domains of complementary specificities could work to more efficiently digest complex polysaccharides [79] (i.e., the two tandem GH43 domains of N1089). In addition, carbohydrate-binding modules were found as part of a number of the selected CAZyme targets, which may help increase the concentration and/or targeting of these enzymes to their substrates [80]. Additionally, polysaccharide linkages within the insoluble NSP may be inaccessible to the target enzymes and/or CM isolates under the designed assay and growth conditions used here. While bacterial CM isolates were able to utilize CM NSP as a sole carbohydrate source, these cultures reached relatively low density when compared to growth on glucose (Figure 5B) or to that of wild-type *B. theta* VPI-5482 on complex polysaccharides, such as pectins [81]. A comprehensive approach involving the use of an enzymatic cocktail with multiple complementary activities may be required to more effectively digest CM NSP for use as livestock feed stocks.

Novel enzymatic activities may also be required for efficient breakdown of CM, tailored to specific linkages found in NSP (Figure 6). With this in mind, two enzymes from CM-utilizing *Bacteroides* isolates, N1089 and K2605, were selected and screened for enzymatic activity. N1089 was discovered to be an arabinofuranosidase and xylosidase, both activities directly relevant to the degradation of arabinosyl- and xylosyl-containing polysaccharides or sidechains found in CM NSP (Figure 1, Figure 2 and Figure 3). Additionally, we were able to successfully demonstrate activity of N1089 on CM AIR. As the genetic construct assayed with the pNP-substrate screen is a full-length enzyme, it is possible that the dual arabinofuranosidase/xylosidase activity seen is the result of one domain of N1089 functioning as an arabinofuranosidase, while the other may possess xylosidase activity. Indeed, the C-terminal module (“b”) of N1089 was found to cluster with α-L-arabinofuranosidases active on arabinan (Figure 6), suggesting that the N-terminal module (“a”) may be responsible for cleaving xylan backbone linkages common to CM plant cell walls (Figure 1 and Figure 3). K2605 was identified by dbCAN as a GH28 and clustered with numerous other endo-polygalacturonases (Figure 6). It is thus likely that the K2605 enzyme is a polygalacturonase, but may accommodate a pNP-galactose substrate in the -1 subsite. Unfortunately, the pNP-α-galacturonide substrate is not commercially available which prevents screening K2605 against this synthetic substrate. Regardless, the galactosidase/galacturonase activity of K2605 may assist with dismantling the homogalacturonan found in CM NSP (Figure 1 and Figure 3). These two enzymes presented here may represent valuable targets for further product development, perhaps as part of a combinatorial strategy to dismantle the complex multitude of polysaccharides present in CM.

Surprisingly, species not commonly associated with complex carbohydrate utilization were also isolated from CM-NSP medium, including *Enterococcus faecalis*, *E. faecium*, *E. cecorum*, and *E. avium* (Figure 5). In all likelihood these microorganisms do not possess the enzymatic machinery required to utilize CM NSPs. Inspection of their genomes revealed that *E. faecium* and *E. avium* possess one GH43 and one GH78 enzyme, while the genomes of *E. cecorum* and *E. faecalis* do not [49]. The genome of *E. faecium* does encode two GH28 enzymes; however, whether these enzymes are secreted and active on CM NSPs is not known. As isolations were performed on rich media after initial minimal media selections, it is possible that these species were able to grow on contaminating monosaccharides in the media or on products generated by neighbouring colonies from *Bacteroides* sp. Importantly, two of the enzymes identified in this study were found to possess N-terminal signal peptides, suggesting that at least some CM NSP-relevant enzymes may be secreted for extracellular function (Figure 6).

## 5. Conclusions

The discovery of enzymes that improve the digestion of feed by livestock has been difficult to achieve in the past. The majority of commercially available enzyme additives have been repurposed for applications in biofuel production [82,83]; thus, they have not been optimized for dietary substrates or to function within the intestine of animals. To address both of these limitations, we have developed an enzyme discovery platform that combines glycomic characterization of complex substrates to inform enzyme selection, selective isolation of bacteria that colonize the cecum of the host species, *in silico* identification of enzymes from sequenced genomes, and biochemical characterization of recombinantly produced target enzymes. In this study, two promising enzymes, N1089 and K2605, were identified from chicken-associated *Bacteroides* spp. that are active on CM polysaccharides. This enzyme discovery platform may provide a new route for informed biocatalyst development that can be adapted to a variety of other plant sources, host species, and industrial applications.

## Figures and Tables

**Figure 1 microorganisms-08-01888-f001:**
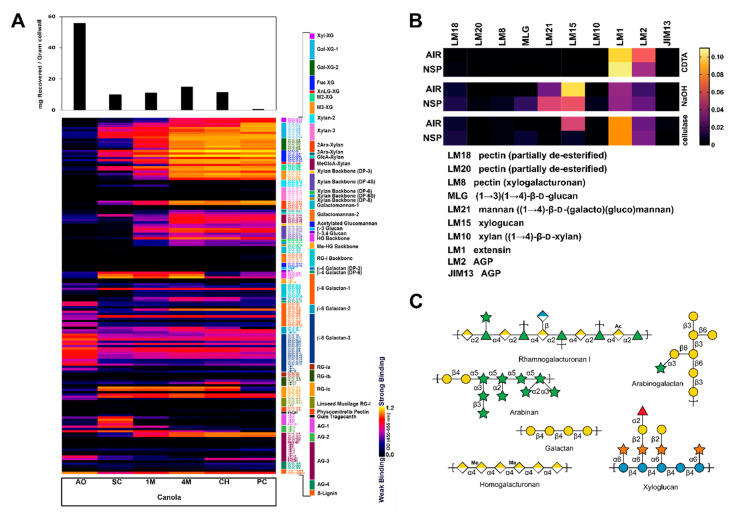
Cell wall glycome profiling of canola meal. Canola meal (CM) plant cell wall extracts were analyzed through an enzyme-linked immunosorbent assay (ELISA)-based screen against a collection of (**A**) 155 plant cell wall or (**B**) 10 glycosidic linkage-specific glycan-directed monoclonal antibodies (mAbs) analysed through MicroArray Polymer Profiling (MAPP). (**A**) Each column represents the different fractions from polysaccharide extractions of the CM plant cell wall: Ammonium oxalate (AO); sodium carbonate (SC); 1 M KOH (1M); 4 M KOH (4M); chlorite (CH); post-chlorite (PC). The colour of each element in the map reflects the strength of the ELISA signal (black = no binding, yellow = strong binding). The antibodies are grouped according to the epitopes that they recognize (bar at the right of the profiles). Each row reflects the binding pattern of a single mAB against the series of CM extracts. The bar graphs above the glycome profiles show the amounts of carbohydrate present in each extract. (**B**) Each column represents the binding pattern of a mAb, as indicated. Each row reflects the different fractions of polysaccharide extractions by CDTA, NaOH, or cellulase treatment for CM alcohol insoluble residue (AIR) or non-starch polysaccharide (NSP). (**C**) Based on the results from the ELISA-based polysaccharide composition screens, schematics were drawn for the dominant polysaccharides found in CM. Monosaccharide symbols follow the SNFG (Symbol Nomenclature for Glycans) system [57].

**Figure 2 microorganisms-08-01888-f002:**
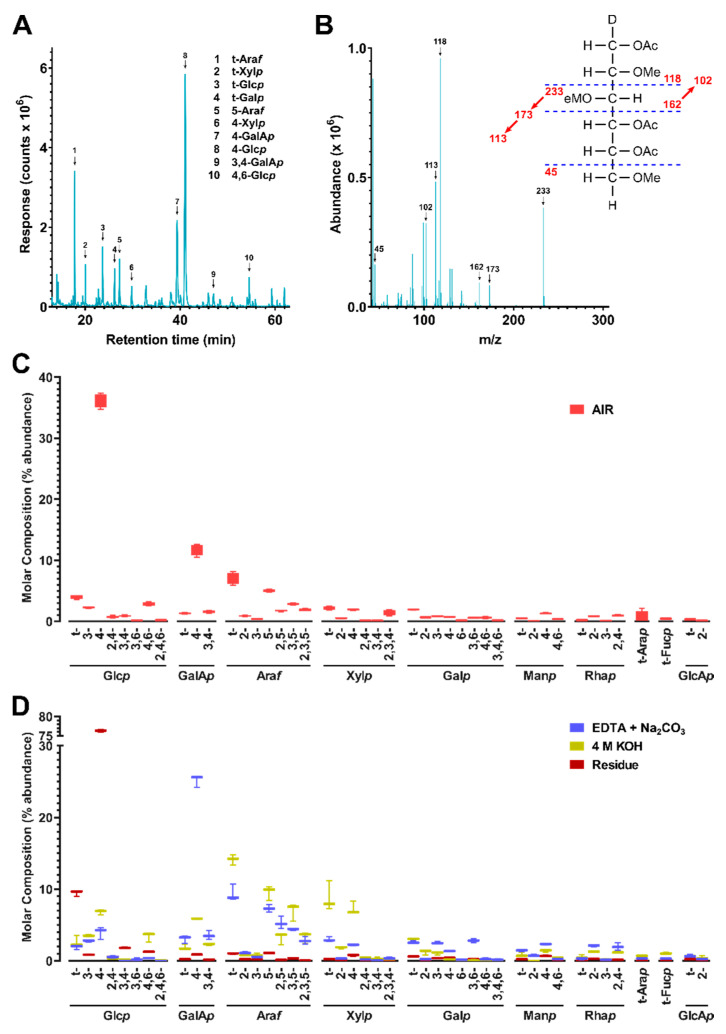
Cell wall linkage composition of cold-pressed canola meal. (**A**) Total ion current (TIC) chromatogram of cold-pressed canola meal (CM) non-starch polysaccharides (NSP) analyzed by PMAA gas chromatography-mass spectrometry (GC-MS). (**B**) Mass spectrum of example PMAA from 4-Glc*p* (β-1,4-linked glucopyranose). Peaks in TIC chromatogram (**A**) were assigned to linkages based on their MS fragmentation patterns and retention times by referring to those of standards. (**C**,**D**) Calculated relative composition of linkages (molar%) from peaks shown in (**A**) for (**C**) CM alcohol insoluble residue (AIR) and (**D**) fractionated extracts of EDTA + Na_2_CO_3_, 4 M KOH, or residue A pre-treatment of NaBD_4_ was conducted to the cell wall before its sequential fractionation in order to increase the extricability of polysaccharides. Four separate experiments were conducted to cell wall and three to each fraction. Values are averages from the separate experiments, presented as bar and whiskers with error bars representing the standard deviation from the mean.

**Figure 3 microorganisms-08-01888-f003:**
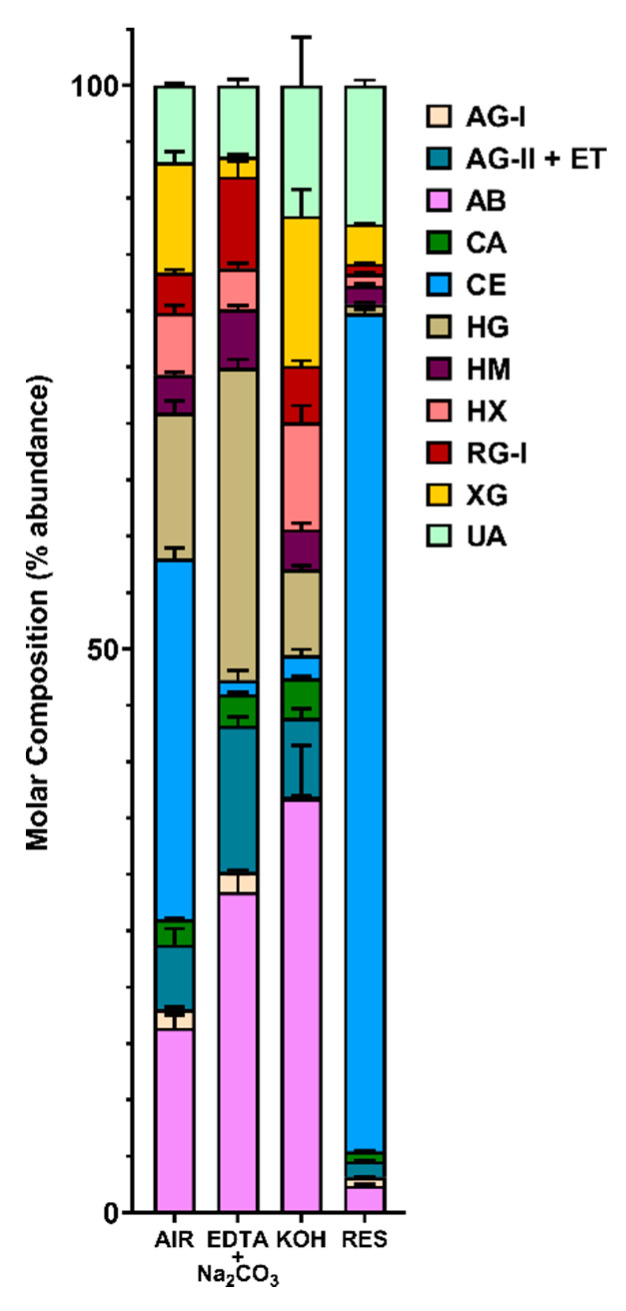
Estimated polysaccharide composition of cold-pressed canola meal cell wall (de-starched AIR) and its three fractions (EDTA + Na_2_CO_3_, 4 M KOH, Residue). The composition of polysaccharides was estimated based on linkage composition data (Figure 2 and Appendix A) by assigning the linkages of uronic acids and neutral sugars to polysaccharides (Table 1; as in [36]). A pre-treatment of NaBD_4_ was conducted to the cell wall before its sequential fractionation in order to increase the extricability of polysaccharides. Four separate experiments conducted to cell wall and three to each fraction. Stacked bar plots are shown, with a mean value from the separate experiments represented as a horizontal bar and bars depicting the standard deviation from the mean. The polysaccharides were assigned as in Table 1: Arabinan (AB), type I arabinogalactan (AG-I), type II arabinogalactan (AG-II) and extensin (ET), heteroxylan (HX), xyloglucan (XG), callose (CA), cellulose (CE), rhamnogalacturonan I (RG-I), homogalacturonan (HG), heteromannan (HM), and unassigned linkages (UA).

**Figure 4 microorganisms-08-01888-f004:**
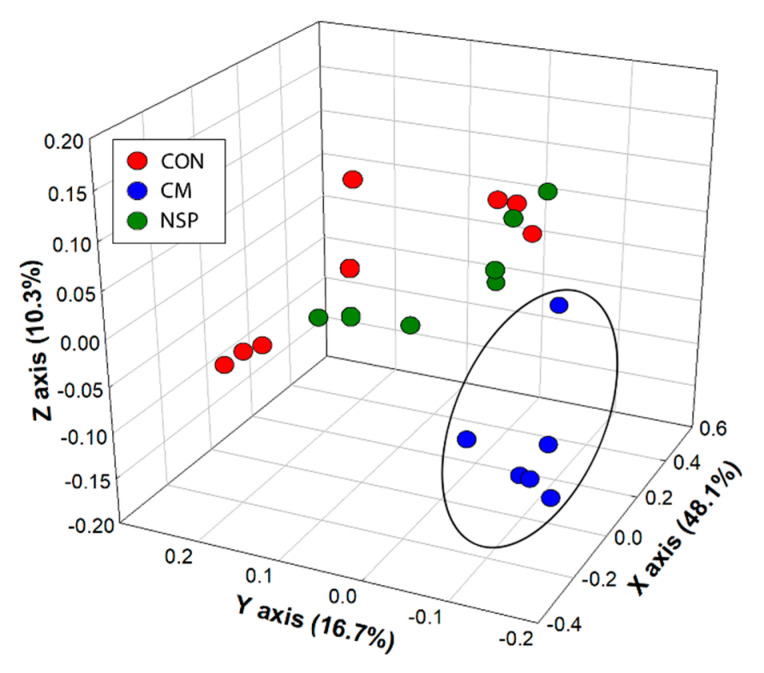
Unweighted UniFrac principal coordinate analysis of cecal communities within mini-bioreactor vessels containing the basal medium (CON), the basal medium with cold press extracted canola meal (CM), and the basal medium with non-starch polysaccharides extracted from canola meal (NSP). Replicate vessels were combined for Illumina characterization of bacterial communities. The ellipsoid illustrates clustering of bacterial communities for the CM treatment, relative to communities for the NSP and CON treatments.

**Figure 5 microorganisms-08-01888-f005:**
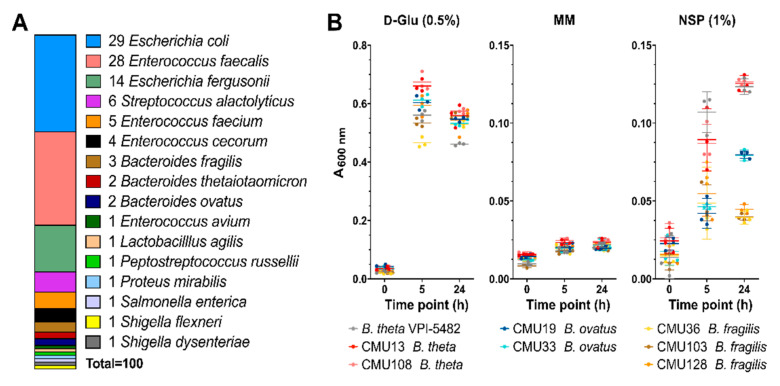
Identification and characterization of bacterial CM NSP degraders. (**A**) 16S rDNA sequencing was performed to identify the 100 bacterial species obtained from isolation plating bioreactor samples on CM NSP. Of those bacterial samples, *Bacteroides* isolate growth was assayed (**B**) in liquid minimal media containing 0.5% D-Glu (positive control), no carbohydrate (MM) (negative control), or 1% CM NSP. Background data was subtracted for media containing no bacterial culture inoculate. Error bars represent the SEM from triplicate data. Liquid culture bacterial isolates are coloured as indicated.

**Figure 6 microorganisms-08-01888-f006:**
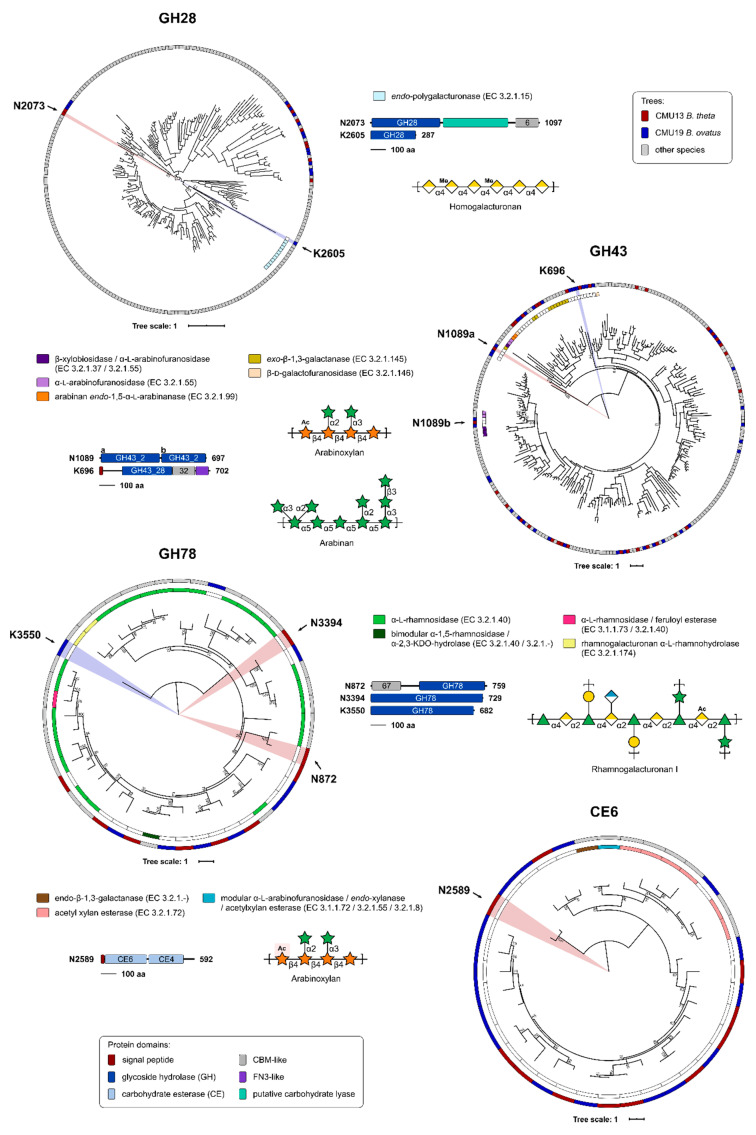
Prediction of enzyme activities and target selection by phylogenetic analyses. Genomes from isolates CM13 (*B. theta*) (red) and CM19 (*B. ovatus*) (blue) were annotated by Prokka [46]. Putative protein sequences were annotated by dbCAN [54] and used as query sequence inputs for SACCHARIS [48] and embedded into phylogenetic trees of characterized GH28, GH43, GH78, and CE6 enzymes. EC number and CAZy database annotated functions are colour-coded as indicated. Target enzymes were chosen based on distally related sequences, represented by novel clades within the phylogenetic trees. Domain boundaries are based on manual curation of predictions by dbCAN [54] and InterProScan [55], with all schematics to scale and colour-coded. The polypeptide length for the open reading frame of each target is shown. Schematics of potential plant cell wall substrates for selected target CAZymes, as identified by EC number, are shown.

**Figure 7 microorganisms-08-01888-f007:**
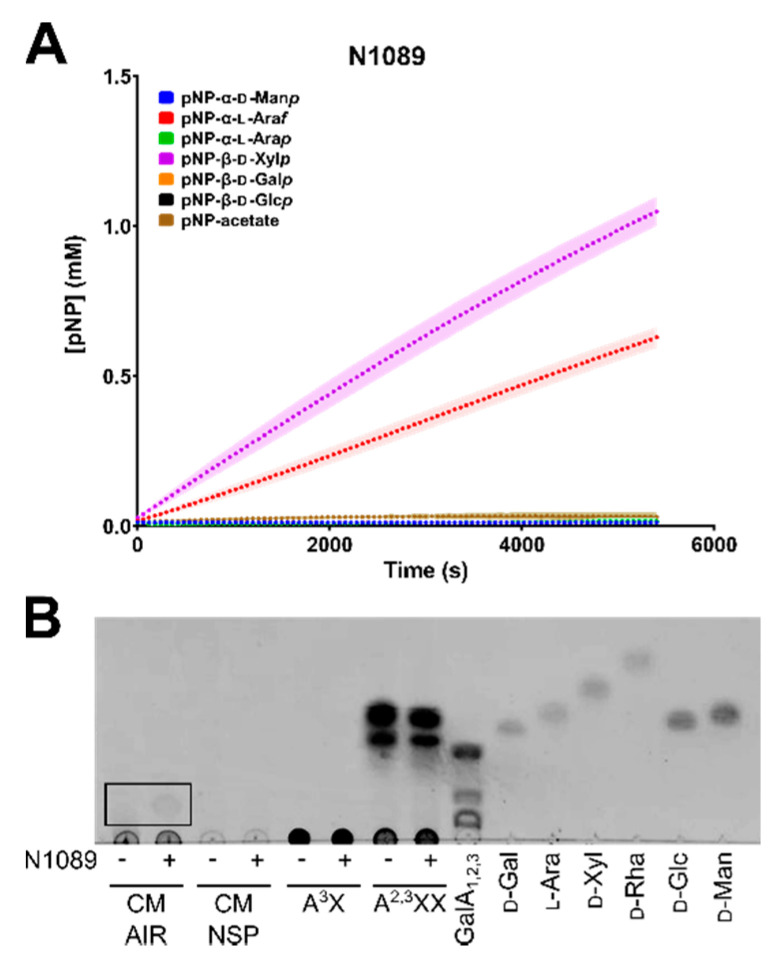
Enzymatic assays for CAZyme activity characterization. Full-length soluble enzymes (**A**) N1089 was incubated with pNP-conjugated substrates, and monitored spectrophotometrically at OD405 nm over a period of 90 min. A standard curve was generated of OD405 nm vs concentration of pNP in order to calculate final product formed by enzyme addition. Plotted points represent an average of three technical replicates, with the standard deviation from the mean shown as a shadow. (**B**) Thin-layer chromatography (TLC) of commercial and synthetic polysaccharides incubated without (−) and with (+) N1089 enzyme. Polysaccharide substrates consisted of: canola meal (CM) alcohol insoluble residue (AIR), CM non-starch polysaccharide (NSP), 3^2^-α-L-arabinofuranosyl-xylobiose (A^3^X), 2^3^,3^3^-di-α-L-arabinofuranosyl-xylotriose (A^2,3^XX), or a galacturonic acid ladder of mono, di, and triose forms (GalA_1,2,3_). Monosaccharide standards were included as controls; D-galactose (D-Gal), L-arabinose (L-Ara), D-xylose (D-Xyl), D-rhamnose (D-Rha), D-glucose (D-Glc), and D-mannose (D-Man). TLC solvent was composed of 2:1:1 butanol:acetic acid:H_2_O and plates were stained with 1% orcinol.

**Table 1 microorganisms-08-01888-t001:** Polysaccharide linkage assignments.

Polysaccharides	Linkage Assignments ^1^
Arabinan (AB)	t-Ara*f*, 5-Ara*f*, 2,5-Ara*f*, 2,3,5-Ara*f*
Type I arabinogalactan (AG-I)	t-Ara*f*, 4-Gal*p*, 4,6-Gal*p*
Type II arabinogalactan (AG-II)	t-Ara*p*, t-Ara*f*, 2-Ara*f*, 3-Ara*f*, t-Rha*p*, t-Fuc*p*, t-Gal*p*, 3-Gal*p*, 6-Gal*p*, 3,6-Gal*p*, 3,4,6-Gal*p*
Extensin (ET)	t-Ara*f*, 2-Ara*f*, 3-Ara*f*, t-Gal*p*
Heteroxylan (HX)	t-Ara*f*, 4-Xyl*p*, 2,4-Xyl*p*, 3,4-Xyl*p*, 2,3,4-Xyl*p*, t-GlcA*p*
Xyloglucan (XG)	t-Xyl*p*, 2-Xyl*p*, t-Fuc*p*, t-Gal*p*, 2-Gal*p*, 4-Glc*p*, 4,6-Glc*p*
Cellulose (CE)	4-Glc*p*
Callose (CA)	3-Glc*p*
Rhamnogalacturonan I (RG-I)	2-Rha*p*, 2,4-Rha*p*, 4-GalA*p*, 3,4-GalA*p*
Homogalacturonan (HG)	t-GalA*p*, 4-GalA*p*
Heteromannan (HM)	4-Man*p*, 4,6-Man*p*, t-Gal*p*, 4-Glc*p*

^1^ The assignments of polysaccharide linkages were according to Pettolino, et al. [36]. No 4-Glc*p* linkage was assigned to cellulose in F_EDTA+Na2CO3_ and F_4MKOH_ according to previously reported results [33].

**Table 2 microorganisms-08-01888-t002:** Whole genome sequencing SPAdes assembly statistics.

Sequencing Statistics	*B. theta* CMU13	*B. theta* CMU108	*B. ovatus* CMU19	*B. ovatus* CMU33	*B. fragilis* CMU36	*B. fragilis* CMU103
# contigs (≥0 bp)	108	139	121	186	136	104
# contigs (≥1000 bp)	53	72	64	71	51	46
# contigs (≥5000 bp)	41	53	54	55	37	39
# contigs (≥10,000 bp)	38	49	47	49	32	33
# contigs (≥25,000 bp)	34	41	44	44	24	27
# contigs (≥50,000 bp)	29	33	36	35	23	25
Total length (≥0 bp)	6,108,120	6,066,561	6,545,746	6,567,743	4,952,122	4,943,672
Total length (≥1000 bp)	6,095,090	6,049,562	6,523,768	6,526,591	4,933,804	4,922,272
Total length (≥5000 bp)	6,069,390	6,004,031	6,501,313	6,492,943	4,894,273	4,904,061
Total length (≥10,000 bp)	6,052,706	5,976,677	6,448,873	6,445,371	4,856,385	4,859,213
Total length (≥25,000 bp)	5,970,234	5,809,415	6,385,940	6,350,457	4,737,700	4,774,286
Total length (≥50,000 bp)	5,803,058	5,512,500	6,081,056	6,013,548	4,702,535	4,702,586
# contigs	58	81	75	88	56	57
Largest contig	939,979	649,114	505,642	473,656	678,395	622,851
Total length	6,099,104	6,055,875	6,532,103	6,538,466	4,937,414	4,929,730
GC (%)	42.84	42.84	41.61	41.61	43	43
N50	231,388	198,325	199,430	206,721	272,224	222,725
N75	126,671	112,416	125,530	125,530	183,065	122,181
L50	8	11	11	11	7	6
L75	16	21	22	21	12	14
# N’s per 100 kbp	6.31	11.18	23.77	13.55	7.13	7.34

Note: Quality assessment statistics were determined without reference by QUAST [45]. All statistics are based on contigs of size ≥ 500 bp, unless otherwise noted (e.g., “# contigs (≥0 bp)” and “Total length (≥0 bp)” include all contigs).

**Table 3 microorganisms-08-01888-t003:** CAZome of bacterial isolates for the major glycoside hydrolase (GH) families involved in degradation of plant cell walls.

CAZy Family	Major Known Activity	*B. theta* VPI-5482	*B. theta* CMU13	*B. theta* CMU108	*B. ovatus* ATCC8483	*B. ovatus* CMU19	*B. ovatus* CMU33	*B. fragilis* CMU36	*B. fragilis* CMU103
**Cellulose- and pectin-active**									
GH28	galacturonases	10	10	10	14	15	15	0	0
GH95	α-L-fucosidase	5	5	5	7	**9**	**9**	4	4
GH88	β-glucouronyl hydrolase	4	3	3	8	6	6	1	1
GH5	cellulase	0	**1**	**1**	5	**7**	**7**	1	1
GH53	endo-1,4-β-galactanase	1	1	1	1	**2**	**2**	0	0
GH10	endo-1,4-β-xylanase	0	0	0	7	3	3	0	0
GH9	endoglucanase	0	0	0	1	1	1	0	0
GH11	xylanase	0	0	0	0	0	0	0	0
GH12	endoglucanase & xyloglucan hydrolysis	0	0	0	0	0	0	0	0
GH26	β-mannanase & xylanase	0	0	0	3	0	0	2	2
GH44	endoglucanase	0	0	0	0	0	0	0	0
GH45	endoglucanase	0	0	0	0	0	0	0	0
GH48	endo-processive cellulases	0	0	0	0	0	0	0	0
GH6	endoglucanase	0	0	0	0	0	0	0	0
GH7	endoglucanase	0	0	0	0	0	0	0	0
GH8	endo-xylanases	0	0	0	0	0	0	0	0
	Total	20	20	20	46	43	43	8	8
**Cell wall elongation**									
GH16	xyloglucanases & xyloglycosyltransferases	3	4	4	4	7	7	4	4
GH74	endoglucanases & xyloglucanases	0	1	1	0	0	0	1	1
GH17	1,3-β-glucosidases	0	0	0	0	0	0	0	0
GH81	1,3-β-glucanase	0	0	0	0	0	0	0	0
	Total	3	5	5	4	7	7	5	5
**Debranching enzymes**									
GH78	α-L-rhamnosidase	6	**9**	**9**	8	**9**	**9**	2	2
GH106	α-L-rhamnosidase	3	**4**	**4**	4	4	4	0	0
GH51	α-L-arabinofuranosidase	4	4	4	4	4	4	2	2
GH23	peptidoglycan lyase	3	**4**	3	3	3	3	3	3
GH33	trans-sialidase	2	2	2	5	5	5	4	4
GH146	β-L-arabinofuranosidase	3	2	2	3	3	3	0	0
GH27	α-galactosidase	5	2	2	3	2	2	2	2
GH77	4-α-glucanotransferase	1	1	1	1	1	1	1	1
GH84	N-acetyl β-glucosaminidase	1	1	1	0	0	0	1	1
GH67	α-glucuronidase	1	0	0	2	2	2	0	0
GH103	peptidoglycan lytic transglycosylase	0	0	0	0	0	0	0	0
GH54	α-L-arabinofuranosidase	0	0	0	0	0	0	0	0
GH62	α-L-arabinofuranosidase	0	0	0	0	0	0	0	0
	Total	29	29	28	33	33	33	15	15
**Oligosaccharide-degrading enzymes**									
GH2	β-galactosidases and other β-linked dimers	32	**34**	**34**	37	32	32	15	15
GH43	arabinases & xylosidases	34	33	33	35	**39**	**39**	10	10
GH92	α-1,2-mannosidase	23	**26**	**26**	18	**20**	**20**	8	8
GH20	β-hexosaminidase	14	13	13	13	11	11	11	11
GH3	mainly β-glucosidases	10	**11**	**11**	21	**23**	**23**	10	10
GH105	unsaturated rhamnogalacturonyl hydrolase	7	**9**	**9**	12	**15**	**15**	0	0
GH18	chitinase	12	9	9	8	**10**	**10**	2	2
GH97	glucoamylase, α-glucosidase, α-galactosidase	10	9	9	12	9	9	4	4
GH29	α-L-fucosidase	9	9	9	7	5	5	9	9
GH130	β-1,4-mannosylglucose phosphorylase	4	**6**	**6**	8	6	6	2	2
GH31	α-glucosidase	6	6	6	12	6	6	4	4
GH35	β-galactosidase	3	**4**	**4**	2	**5**	**5**	3	3
GH32	invertase, endo-inulinase	4	4	4	2	**3**	**3**	2	2
GH38	α-mannosidase	2	**4**	**4**	1	1	1	1	1
GH127	β-L-arabinofuranosidase	3	3	3	1	**4**	**4**	5	5
GH13	α-amylase	8	2	2	5	2	2	2	2
GH141	α-L-fucosidase, xylanase	2	2	2	2	2	2	1	1
GH27	α-galactosidase	5	2	2	3	2	2	2	2
GH137	β-L-arabinofuranosidase	1	2	2	1	1	1	1	1
GH42	β-galactosidase	1	1	1	1	**3**	**3**	0	0
GH138	α-galacturonidase	1	1	1	1	1	1	0	0
GH139	α-2-O-Me-L-fucosidase	1	1	1	1	1	1	0	0
GH142	β-L-arabinofuranosidase	1	1	1	1	1	1	1	1
GH143	2-keto-3-deoxy-D-lyxo-heptulosaric acid hydrolase	1	1	1	1	1	1	0	0
GH57	α-amylase	1	1	1	1	1	1	1	1
GH147	β-galactosidase	0	0	0	1	**2**	**2**	0	0
GH1	β-glucosidase and many other β-linked dimers	0	0	0	0	0	0	0	0
GH39	β-xylosidase	0	0	0	0	0	0	0	0
GH52	β-xylosidase	0	0	0	0	0	0	0	0
GH94	cellobiose phosphorylase	0	0	0	0	0	0	0	0
	Total	195	194	194	207	206	206	94	94

Underlined and bolded values highlight CAZy families with more members in the CM isolates than the wild-type strains (*B. theta* VPI-5482 and *B. ovatus* ATCC8483).

**Table 4 microorganisms-08-01888-t004:** CAZome of bacterial isolates for the major carbohydrate esterase (CE) and polysaccharide lyase (PL) families involved in degradation of plant cell walls.

CAZy Family	Major Known Activity	*B. theta* VPI-5482	*B. theta* CMU13	*B. theta* CMU108	*B. ovatus* ATCC8483	*B. ovatus* CMU19	*B. ovatus* CMU33	*B. fragilis* CMU36	*B. fragilis* CMU103
**Polysaccharide lyases**									
PL22	oligogalacturonate/oligogalacturonide lyase	0	**3**	**3**	0	**3**	**3**	4	4
PL9	pectate lyase	2	2	2	3	1	1	0	0
PL10	pectate lyase	1	1	1	1	1	1	0	0
PL26	rhamnogalacturonan exolyase	1	1	1	1	1	1	0	0
PL1	pectate lyase	5	0	0	9	0	0	0	0
PL2	pectate lyase	0	0	0	0	0	0	0	0
PL3	pectate lyase	0	0	0	0	0	0	0	0
PL4	rhamnogalacturonan endolyase	0	0	0	0	0	0	0	0
	Total	9	7	7	14	6	6	4	4
**Carbohydrate esterases**									
CE6	acetyl xylan esterase	1	**11**	**11**	3	**17**	**17**	3	3
CE12	pectin acetylesterase	2	**5**	**5**	8	7	7	0	0
CE4	chitin deacetylase	1	**4**	**4**	2	**5**	**5**	2	2
CE8	pectin methylesterase	2	2	2	6	6	6	0	0
	Total	6	22	22	19	35	35	5	5

Underlined and bolded values highlight CAZy families with more members in the CM isolates than the wild-type strains (*B. theta* VPI-5482 and *B. ovatus* ATCC8483).

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
