# Peer review of "Combinatorial Glycomic Analyses to Direct CAZyme Discovery for the Tailored Degradation of Canola Meal Non-Starch Dietary Polysaccharides"

_microorganisms, 2020, doi:10.3390/microorganisms8121888_

Round 1

Reviewer 1 Report

In “Combinatorial glycomic analyses to direct CAZyme discovery for the tailored degradation of canola meal non-starch dietary polysaccharides,” Low, et al, use a comprehensive carbohydrate chemistry approach first to identify the diversity of carbohydrate structures that are abundant in canola meal non-starch polysaccharides extracted in different ways, revealing antibody reactivity, neutral monosaccharide and linkage structures that were used to estimate the structures present in canola meal extracts. They then use those NSP extracts to select for chicken microbiota that are capable of degrading these polymers in mini-bioreactors, identifying multiple strains of Bacteroides spp. thetaiotaomicron, ovatus, and fragilis. They then sequence isolate genomes and predict CAZymes of interest for potential enzymatic conversion of NSPs for improving chicken efficiency on this feedstock and demonstrate in vitro activity of two candidates using heterologous expression and in vitro enzyme assays. The manuscript is very well written, and the conclusion that the two enzymes identified do, indeed, possess the predicted enzymatic activities is well-supported. I have only moderate and minor concerns.

Moderate concerns:

  1. It’s unclear to me why so many enterobacteria were able to grow on the CM-NSP agar medium, as these organisms are very unlikely to access the polymers detected in the mix. Perhaps consuming tryptone as a C source? Cross-feeding of sugars from neighboring colonies? The authors should comment on this unexpected result.
  2. Figure 6 – the phylogenetic trees are very small and hard to read. There is plenty of white space in this figure; these should be expanded so that they are much easier to interpret. To increase size while minimizing conflict between the radial diagrams, it would be possibly helpful to alternate the sides of trees/legends, domain diagrams, and polysaccharide models.
  3. Figure 7 – this legend does not seem right, or else the figure isn’t correct – I only see panel A and B, and it seems like only N1089. B is TLC, not C (as in the legend). Please correct.
  4. The Discussion section should be improved to provide more context for the two enzymes identified in this study. How does this work improve the broader understanding of these gene families in diverse gut microbiomes? Although the goal of understanding feed additives is good, lack of comment on this broader impact seems like a missed opportunity.

Minor comments:

L152 – Missing “the” before “soluble”

L215 – Exact primer sequences employed should be provided here (so that readers do not need to hunt down the reference)

L220 – How were samples indexed?

L234 – Suggest hyphenating “Bray-Curtis” throughout

L320 – BioTek what?

L416 – “plants lack”

L468 – remove comma before “Bacteroides”

L515 – “CAZome” appears to be a proliferation of not very helpful “-ome” words. It’s much better in my mind to use a term that all will understand. Perhaps “CAZyme-encoding gene complements?”

L648 – Suggest “rich” rather than “abundant” – either that, or flip the order “pectins, XG, and AX are abundant in CM NSP”

Author Response

Reviewer 1:

In “Combinatorial glycomic analyses to direct CAZyme discovery for the tailored degradation of canola meal non-starch dietary polysaccharides,” Low, et al, use a comprehensive carbohydrate chemistry approach first to identify the diversity of carbohydrate structures that are abundant in canola meal non-starch polysaccharides extracted in different ways, revealing antibody reactivity, neutral monosaccharide and linkage structures that were used to estimate the structures present in canola meal extracts. They then use those NSP extracts to select for chicken microbiota that are capable of degrading these polymers in mini-bioreactors, identifying multiple strains of Bacteroides spp. thetaiotaomicron, ovatus, and fragilis. They then sequence isolate genomes and predict CAZymes of interest for potential enzymatic conversion of NSPs for improving chicken efficiency on this feedstock and demonstrate in vitro activity of two candidates using heterologous expression and in vitro enzyme assays. The manuscript is very well written, and the conclusion that the two enzymes identified do, indeed, possess the predicted enzymatic activities is well-supported. I have only moderate and minor concerns.

Moderate concerns:

  1. It’s unclear to me why so many enterobacteria were able to grow on the CM-NSP agar medium, as these organisms are very unlikely to access the polymers detected in the mix. Perhaps consuming tryptone as a C source? Cross-feeding of sugars from neighboring colonies? The authors should comment on this unexpected result.

We agree that this is unusual to have isolated so many Enterococcal species with CM NSP. We have hypothesized that this may occur from extracellular degradation of CM NSPs, producing mono/oligosaccharides accessible to species incapable of producing the variety of enzymes required, and that these species were indeed isolated based on cross-feeding of sugars from neighbouring colonies. We have added a paragraph to the Discussion to comment on these surprising results (lines 704-720).

  1. Figure 6 – the phylogenetic trees are very small and hard to read. There is plenty of white space in this figure; these should be expanded so that they are much easier to interpret. To increase size while minimizing conflict between the radial diagrams, it would be possibly helpful to alternate the sides of trees/legends, domain diagrams, and polysaccharide models.

We have made edits to Figure 6 to address these concerns. The size of the phylogenetic trees has been increased, and the layout of the figure changed to fit.

  1. Figure 7 – this legend does not seem right, or else the figure isn’t correct – I only see panel A and B, and it seems like only N1089. B is TLC, not C (as in the legend). Please correct.

The figure caption has been corrected.

  1. The Discussion section should be improved to provide more context for the two enzymes identified in this study. How does this work improve the broader understanding of these gene families in diverse gut microbiomes? Although the goal of understanding feed additives is good, lack of comment on this broader impact seems like a missed opportunity.

Within our added paragraph to the Discussion (lines 704-720), as described above, we have added comments to discuss the impact of extracellular expression of some of these enzymes identified in this study and how that may affect communal species. For the two specific enzymes that were able to express and purify here, N1089 and K2605, we think it may be premature to speculate about the broader context of these enzymes. Future research will be necessary to investigate if these enzymes are  indeed upregulated in the chicken cecal microbiota, and the impact these may have in the digestion of CM NSP.

Minor comments:

L152 – Missing “the” before “soluble”

Corrected.

L215 – Exact primer sequences employed should be provided here (so that readers do not need to hunt down the reference)

Added.

L220 – How were samples indexed?

Added “with a dual-indexing strategy”, as defined by Kozich et al., ref 38. (line 214)

L234 – Suggest hyphenating “Bray-Curtis” throughout

Corrected. Lines 237 and 477.

L320 – BioTek what?

Corrected. “BioTek Instruments”

L416 – “plants lack”

Corrected.

L468 – remove comma before “Bacteroides”

Corrected.

L515 – “CAZome” appears to be a proliferation of not very helpful “-ome” words. It’s much better in my mind to use a term that all will understand. Perhaps “CAZyme-encoding gene complements?”

“CAZome” is a well established term in the CAZyme field and is increasingly being used in the literature to describe the total CAZyme content of an organism. We have defined this term as such within this manuscript (line 281-282), and wish to continue to use this to be consistent with other articles within the field.

L648 – Suggest “rich” rather than “abundant” – either that, or flip the order “pectins, XG, and AX are abundant in CM NSP”

Corrected to “rich”.

Reviewer 2 Report

The article presents an excellent work that investigated microbial enzymes specific for depolymerization of canola meal non-starch polysaccharides.

Comments for the Authors:

In the abstract (L 33-36) and introduction (L 61 -64) the authors suggested using enzymes from herbovores/ruminants to improve the digestion of CM. However, in the Materials and Methods, only ceca digesta was utilized as the source of microbiota. This should be clarified as it may be confusing.

L 173-175: How many broiler breeders were euthanized? Please provide more information regarding the euthanasia and the time period after euthanasia when digesta were collected.

Author Response

Reviewer 2:

The article presents an excellent work that investigated microbial enzymes specific for depolymerization of canola meal non-starch polysaccharides.

Comments for the Authors:

In the abstract (L 33-36) and introduction (L 61 -64) the authors suggested using enzymes from herbovores/ruminants to improve the digestion of CM. However, in the Materials and Methods, only ceca digesta was utilized as the source of microbiota. This should be clarified as it may be confusing.

We have added a clarification to line 63 on our use of chicken cecal samples: “or enriching the abundance of limiting enzyme activities found within the chicken microbiota”.

L 173-175: How many broiler breeders were euthanized? Please provide more information regarding the euthanasia and the time period after euthanasia when digesta were collected.

We have added clarification to the text (clarification underlined): (lines 173-175) Digesta were harvested within 30 min of death from the ceca of three healthy ca. 6-month-old mature male broiler breeders obtained from a local producer (i.e. humanely culled by the producer according to industry standards and transported to the Lethbridge Research and Development Centre immediately after euthanization).
